# A Human-in-the-Loop Approach for Personal Knowledge Graph Construction from File Names

Markus Schröder, Christian Jilek, and Andreas Dengel

[1] Smart Data & Knowledge Services Dept., DFKI GmbH, Kaiserslautern, Germany
[2] Computer Science Dept., TU Kaiserslautern, Germany
{markus.schroeder, christian.jilek, andreas.dengel}@dfki.de

**Abstract.** Knowledge workers' personal and work related concepts (e.g. persons, projects, topics) are usually not sufficiently covered by knowledge graphs. Yet, already handmade classification schemes, prominently folder structures, naturally mention several of their concepts in file names. Thus, such data could be a promising source for constructing personal knowledge graphs. However, this idea poses several challenges: file names are usually noisy non-grammatical text snippets, while folder structures do not clearly define how concepts relate to each other. To cope with this semantic gap, we include knowledge workers as humans-in-the-loop to guide the building process with their feedback. Our semi-automatic personal knowledge graph construction approach consists of four major stages: domain term extraction, ontology population, taxonomic and non-taxonomic relation learning. We conduct a case study with four expert interviews from different domains in an industrial scenario. Results indicate that file systems are promising sources and, combined with our approach, already yield useful personal knowledge graphs with moderate effort spent.

**Keywords:** Knowledge Graph Construction · Personal Knowledge Graph · Human-in-the-Loop · File System

## 1 Introduction

Knowledge graphs (KGs) have become a popular technology to support knowledge workers in various applications (for a survey see [8]). Since such KGs are constructed from domain-specific document corpora, personal concepts of knowledge workers in these domains are usually not sufficiently covered. To fill this gap, there is the emerging concept of Personal Knowledge Graphs (PKGs) which focus on resources users are personally related to (also in their professional life). The population and maintenance of such graphs is still an open research question [1], especially, when knowledge is not modeled yet (cold start problem). Various sources in a user's personal information sphere may be worth considering to kick-start a population [12].

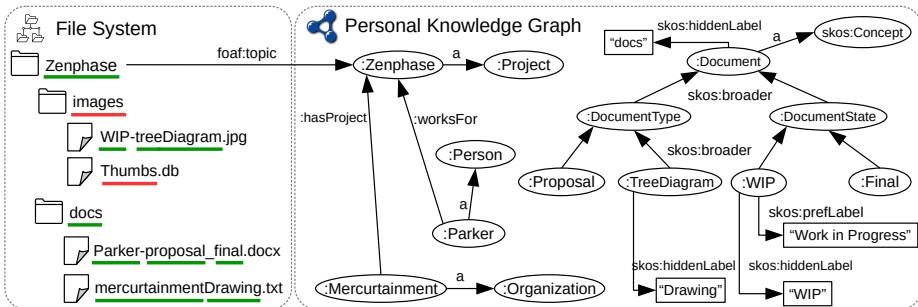

Fig. 1: A file system (left) with file names containing relevant words (green) and irrelevant words (red). They form a personal knowledge graph (right) with non-taxonomic and taxonomic relations. Due to readability, some edges are omitted.

When users self-organize diverse documents in daily business, they often manage them in a form of classification schema, prominently in file systems [4]. Here, documents are hierarchically arranged and freely named according to aspects such as projects, organizations, persons, topics and task-related concepts. In file and folder names such concepts are typically mentioned in order to let users guess their contents. Because file systems allow to name them mostly free[3], users tend to label them with their own vocabulary which can contain technical terms, made-up words or even puns [2]. Thus, we hypothesize that file names could be a promising source for constructing PKG.

This idea poses several challenges due to the nature of the data source. Literature already showed that users have a large variety of file naming strategies [5,3]. File names are usually short ungrammatical (sometimes noisy) text snippets and contain differently ordered and concatenated keywords. These circumstances make it difficult to discover and extract relevant named entities from them. Besides labeling, users can also assemble files in hierarchically structured folders [14]. Yet, this "folder contains file" structure typically does not explicitly define how named entities relate to each other.

To give a visual example, Figure 1 depicts a small file system (left) and a possible personal knowledge graph (right). Because some keywords in the file names are too general (*images*) or have a technical meaning (*Thumbs*), they may be irrelevant for the user (underlined in red). Relevant keywords (green) become resources in the PKG, while a `foaf:topic` property keeps track in which file resource it is mentioned (only one is shown due to readability). Named individuals (*Zenphase*, *Parker*, *Mercurtainment*) are assigned to their classes (*Project*, *Person*, *Organization*) and are connected meaningfully (`:hasProject`, `:worksFor`). The remaining ones are rather abstract ideas and thus become `skos:Concept`s according to the Simple Knowledge Organization System (SKOS). A taxonomy tree is formed (top-right side) by adding broader concepts (`:DocumentType`, `:DocumentState`). Since *WIP* is an abbreviation, its `skos:prefLabel` contains

---
[3] Restricted only by illegal characters and maximum file name length.

the long form. Synonyms and other spellings are captured in `skos:hiddenLabel`s: for the user the term *Drawing* is synonym to *treeDiagram* and *docs* in file names indicate the concept *Document*. Due to the lack of space, labels and some other properties are not visualized.

In this paper, we present a semi-automatic personal knowledge graph construction approach which is able to build such a graph from a classification schema, in this case, a file system and expert feedback. A graphical user interface (GUI) assists a knowledge engineer (KE) in performing several tasks during construction: the discovery of concepts in file names, ontology population of concepts and learning of taxonomic as well as non-taxonomic relations. In an interview setting an expert can describe his or her personal view on their files to the KE who translates the explanations in suitable knowledge graph statements using the GUI. To reduce the manual effort for the KE, we make use of machine learning models which learn from feedback and predict new statements during usage. This proposed method yields several research questions (RQs), for which first answers are reported in this work.

- RQ1: *Are file systems promising sources for knowledge graph construction?*
- RQ2: *Can our system suggest helpful statements during usage?*
- RQ3: *How efficient is the construction in our approach?*

The rest of this paper is structured as follows: related approaches are covered in the next section (Sec. 2). This is followed by the presentation of our approach in Section 3 and a prototypical implementation in Section 3.6. The above research questions are then addressed in a case study with expert interviews in Section 4. Section 5 closes the paper with a conclusion and future work.

## 2   Related Work

To personally assist knowledge workers in their tasks, knowledge services benefit from personal information models about users [12]. For building such a model, personal concepts can be acquired from various texts in a user's personal information sphere [13]. Thus, folder structures could be useful for this purpose which is also investigated by other related works.

Magnini et al. [10] as well consider hierarchical classifications and analyze the implicit knowledge hidden in the labeled nodes. They use logic formulas expressed in description logic and word senses discovered and disambiguated in labels to make knowledge explicit. Contextual interpretations such as implicit disjunctions and negations are performed by exploiting the hierarchy. In contrast to our work, their goal is the definition of an ontology with classes and properties (TBox) by relying on external language repositories containing word senses. For us the usage of such resources is limited, since word senses of personal concepts (like projects) are usually not contained. Moreover, they present a fully automatic approach without integrating domain experts in cases where labels do not match with any entry in dictionaries.

More closely related is the work about knowledge extraction from classification schemes by Lamparter et al. [9]. Following the same motivation, the authors would like to acquire explicit semantic descriptions from legacy information such as local folder structures. To archive this, their processing pipeline include the identification of concept candidates, word sense disambiguation, taxonomy construction and identification of non-taxonomic relations. They distinguish ontology and instance layer by checking with dictionaries if terms are rather general (concepts) or specific (instances). In our approach, we only consider instances, but classify general ideas as `skos:Concept`s (e.g. *Diagram*). They also build a taxonomy by utilizing hyponym and hyperonym information. In case of non-taxonomic relations, the work reuses domain-specific ontologies, while the classification hierarchy as well as its labels are consulted to guess appropriate relations. Our procedure is similar, but additionally considers user feedback to train machine learning models in order to predict such relations.

In conclusion, to the best of our knowledge, there is no approach like ours that constructs personal knowledge graphs from folder structures and at the same time includes experts with their feedback.

## 3   Approach

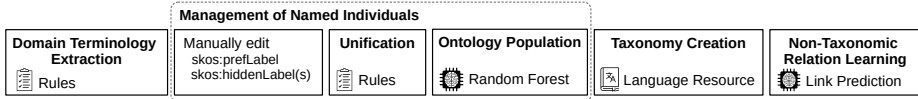

Fig. 2: Components of our approach from left to right.

Our approach enables knowledge engineers (KEs) to construct personal knowledge graphs from a classification schema, for example, a folder structure as shown in Figure 1. In this process, we support them in four tasks which are depicted in Figure 2 and explained in individual sections: Domain Terminology Extraction (Section 3.2), Management of Named Individuals (Section 3.3), Taxonomy Creation (Section 3.4) and Non-Taxonomic Relation Learning (Section 3.5). During modeling using a dedicated GUI (Section 3.6) the KE is assisted by an artificial intelligence (AI) system which proactively makes statements on its own. For ontology population and non-taxonomic relations, machine learning models predict statements. To correctly store and distinguish these assertions, we first designed an appropriate data model.

### 3.1   Knowledge Graph Model

Our knowledge graph model is an RDF graph consisting of statements in the form of subject-predicate-object triples. However, in our scenario, we have to store additional feedback information for each statement. We consider exactly two agents in our system who are able to give feedback about statements: a knowledge engineer (KE) and an artificial intelligence (AI). Both contribute to

the same personal knowledge graph with assertions which can be true, but also false (negative statement). To keep track about the provenance, we store the following meta data for each statement: (a) which agent stated it, (b) the date and time it was stated, (c) how is the statement rated (true, false or undecided) and (d) how confident is the agent (a real value between 0 and 1). Additionally, we use `foaf:topic`-statements to state that a classification schema node (subject) mentions a certain knowledge graph resource (object) (see an example in Figure 1). Regarding the rating, since natural intelligence is usually more reliable than an artificial one, the KE always outvotes suggestions from the AI. Yet, assertions of the AI are assumed to be true as long as the KE does not disagree.

### 3.2 Domain Terminology Extraction

Our extraction method uses heuristics to make a first guess for relevant terms in the user's domain. Since word boundaries are often not evident in rather messy file names, we tokenize their basenames (without considering file extensions) by character type and camel case. In addition, the acquired tokens are rated based on some simple rules: stop words and tokens containing a single letter or only symbols are negatively rated. This also applies for tokens which only contain digits, except they look like years (e.g. $n \in [1980, 2030]$). Applying these rules, the following example is tokenized (indicated by a pipe symbol '|') and rated (indicated by color) in the following way: `WIP|__|for|2007|-|tree|Diagram|!|(|28|)|A|.jpg`. Thus, the rules let us assume that the tokens *WIP*, *2007*, *tree* and *Diagram* are relevant. In case of multi-word terms, the KE is able to merge separated tokens to a single term again, like for the latter two (i.e. *Tree Diagram*).

After adjusting the rating according to feedback from a domain expert, other occurrences of accepted terms are automatically searched using a regular expression, since they may occur in a classification scheme more than once. If the term contains multiple words, we also search for all possible word concatenations using the separators "-" (minus), "_" (underscore), " " (space) and also no separator at all. To give an example, for the term *treeDiagram* our system also checks the variations *tree-Diagram*, *tree_Diagram* and *tree Diagram*. Finally, the collected term variations are associated with a named individual (i.e. `owl:NamedIndividual` according to OWL).

### 3.3 Management of Named Individuals

After retrieving all found term variations $T$, we have to decide if they (a) resemble an already existing named individual or (b) define a new one. Regarding the first case, each newly discovered term may be a variation that refers to an already created named individual. Thus, we calculate the Jaccard similarity coefficient [7] between the terms $T$ and the candidates' labels $L$. A named individual is picked which has the highest overlap between its labels and the given terms. If we cannot find such a resource above a sufficient similarity threshold, a new one is created. The longest term is used to give the resource a preferred label (`skos:prefLabel`) after some conversions are performed: German umlaut spellings are corrected (e.g.

"ae" → "ä"), underscores are replaced with spaces, if available a lemma version is used (*diagrams* → *diagram*) and proper case is applied (*Tree Diagram*). The remaining terms form the named individual's synonym and differently spelled labels (`skos:hiddenLabel`). In both cases, we keep track in which file resource the named individuals are mentioned by using a `foaf:topic`-relation.

**Unification.** If two or more named individuals have the same meaning, we can unify them to one resource. This is done by correctly substituting URIs and at the same time removing the source triples. The AI automatically detects potential individuals with the same meaning by looking at their labels and applying some rules: it checks for hidden labels if they overlap or if there is a prefix or postfix dependency, while preferred labels are compared with the Levenshtein distance and token-based equality. For example, for the following label pairs our procedure would suggest that their individuals are equal: ("Peter Parker", "Parker Peter"); ("Tree Diagram", "Diagram") and ("diagram", "diagramm").

**Ontology Population.** The KE manually create ontology classes and type named individuals with them. To support the KE in this assignment, a random forest model [6] is trained with positive examples from feedback to be able to predict classes for individuals without a type. In order to acquire training features, we follow a gazetteer-based embedding technique by looking up words from several gazetteer lists in preferred labels of named individuals. Remaining characters are counted per character class such as spaces, quotes and digits. The coverage proportions of words and characters in the label serve as the final feature vector. To give some examples, "Tree Diagram 27" receives the vector $v_1 = (\text{English Noun} = 0.73, \text{Space} = 0.13, \text{Digit} = 0.13)$, while "WIP" has $v_2 = (\text{Uppercase Letter} = 1.0)$. Having such feature vectors, the random forest model is able to learn decision trees which predict the same type for named individuals having preferred labels very similar in content. For instance, since the individual *Tree Diagram 27* is assigned to `skos:Concept` and another individual *Diagram 3* has a similar feature vector, our model predicts the same class for it.

### 3.4   Taxonomy Creation

Our intended taxonomy uses broader and narrower relations to structure concepts (`skos:Concept`) found in file names according to the Simple Knowledge Organization System (SKOS). Since we see these concepts as leafs in a taxonomy tree, our motivation is to find broader concepts for them. For this, our approach utilizes a language resource of synsets and hypernym relations. The concepts in the PKG are mapped via their labels to synsets of the lexical-semantic net. By traversing hypernym relations for all found synsets, two or more of them may share the same ancestor along their hypernym paths. If the average distance from synsets to ancestor is below a configurable threshold, it is suggested as a broader concept for them. This constraint avoids the recommendation of too general concepts (e.g. near the root node). To give an example, given the hypernym paths *diagram* → *depiction* and *timetable* → *overview* → *depiction*, our procedure would suggest the broader concept *depiction* for both leafs. Of course the KE may at any time create concepts manually and link them accordingly. Besides

such taxonomic relations, our system also considers non-taxonomic ones between instances.

### 3.5   Non-Taxonomic Relation Learning

To predict non-taxonomic relations, we perform link prediction by training a model on positive examples from feedback and by exploiting the structure of the classification schema (CS). Our idea is that the same non-taxonomic predicate could be suggested between other resources (subjects and objects) which have a similar neighborhood in the CS. For this, we only consider class instances which are named individuals that have been assigned to an ontology class. Since instances are annotated on files via a `foaf:topic`-relation, we know in which places of the CS they are mentioned. This annotated CS needs to be transformed into an undirected graph of connected instances to perform link prediction on it. We make an edge from an instance $i$ mentioned in a given node to another instance $j$, whenever $j$ is mentioned in the (a) node itself, (b) the node's parent, (c) one of the node's children or (d) one of the node's siblings (i.e. children of parent). In other words, instances are connected in the graph if they are closely mentioned in the CS. With the given graph, we are able to calculate local similarity measures for links (for a survey see [11, Table 1]). Values of the calculated measures form feature vectors in a training set. The test set is acquired by iterating over all possible combinations of instances and properties by using their domain and range information as a filter. A promising triple in the test set is expected when we calculate a small euclidean distance (below a given threshold) between its test vector and a training vector.

### 3.6   Prototypical Implementation

To test our approach in a case study, we implemented a prototype. A demo video[4] and its source code[5] are publicly available. To assist the KE in entering feedback and constructing the PKG, a graphical user interface (GUI) in form of a web application is provided (see Figure 3). Throughout the interface, we make heavily use of thumbs-up and thumbs-down buttons as well as green and red colored elements to visualize positive and negative feedback (true and false assertions). The three-column layout presents tabs for individual components which give dedicated views for the tasks we have discussed.

A typical **Explorer** view (top left) lists containing files of a currently browsed folder (`/User/Downloads`). The view presents for each file (from top to bottom) its file name, rated terms from the file name and annotated named individuals. To distinguish individuals from terms the well-known hashtag symbol is added to their preferred labels. In a separate **Named Individuals** view in the top middle, we itemize them together with their type. Two side-by-side views enable a Drag&Drop mechanism on individuals to let the KE define triples with a selected

---

[4] `https://www.dfki.uni-kl.de/~mschroeder/demo/kecs`
[5] `https://github.com/mschroeder-github/kecs`

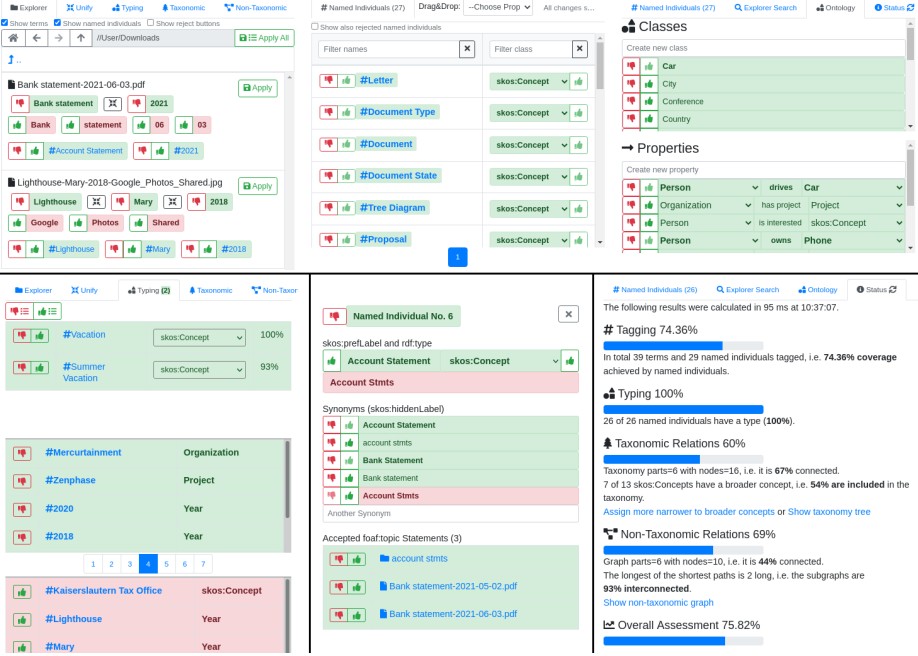

Fig. 3: Our graphical user interface in a three-column layout with many feedback possibilities and components (top). Dedicated components are provided to preform certain tasks (bottom).

predicate (drop-down list in the middle). On the top right, classes and properties can be manually created, renamed and rated in an **Ontology** view. For each property, domain and range classes can be defined too. In separate tabs (bottom left) our GUI also presents suggestions for **Unification**, **Typing**, **Taxonomic** and **Non-Taxonomic** Relations (the screenshot shows an opened Typing tab). A list of proposals from the AI can be reviewed by the KE, who can accept or reject them individually or in bulk. Decisions are shown below and can always be undone in either way. In a **detail view** (bottom middle), the KE is able to change a selected individual's preferred label, type, hidden labels and file attachment. A **Status** view (bottom right) visualizes the current PKG construction state in four sections: the progress in tagging, typing, taxonomy tree and non-taxonomic graph as well as an overall assessment score. These estimations give hints to the KE where more feedback from the expert is necessary.

## 4   Case Study: Expert Interviews

A case study was conducted with expert interviews in which personal knowledge graphs (PKGs) were built with their feedback. The setup for these interviews is covered in Section 4.1. This is followed by a detailed description of all collected

Table 1: Four datasets with their meta data which are used in interviews with four experts.

| Dataset | Expert | Branches | Leafs | Max. Depth | Avg. Depth | Avg. Name Length |
|---------|--------|----------|-------|------------|------------|------------------|
| SS1 | E1 | 103 | 198 | 3 | $2.98 \pm 0.16$ | $8.84 \pm 9.86$ |
| FS1 | E2 | 25,988 | 95,760 | 17 | $9.49 \pm 1.93$ | $23.30 \pm 16.88$ |
| FS2 | E3 | 8,939 | 64,571 | 17 | $9.18 \pm 1.68$ | $32.43 \pm 16.77$ |
| FS3 | E4 | 54,933 | 325,476 | 22 | $10.08 \pm 2.22$ | $24.24 \pm 14.57$ |

results (Section 4.2) which are then discussed with regard to our stated research questions (Section 4.3).

## 4.1    Expert Interview Setup

Since our institute has industry projects with several departments of a large power supply company, we had the great opportunity to get in contact with four individual experts from four departments (guideline management, property management, license management and accounting). Three of them work separately on individual shared drive file systems (FS), while one primarily manages spreadsheet (SS) data. Before the interviews, we received dumps of their data which are listed in Table 1. For each dataset an expert (E) is assigned and meta data about the asset is presented.

Since spreadsheets may also contain work related concepts, but are not a form of classification schema, we had to convert the SS1 dataset to a tree structure in the following way. Table names become root folders, while column names are added as their subfolders. In the subfolders, we add files with distinct names from the column's rather short cell values. This way, potential work related concepts could be contained in this generated classification schema.

Our system automatically captures several data points during usage. To reproduce the construction process, we keep a history of all stated assertions with their meta data as described in Section 3.1. By observing GUI inputs including mouse clicks, Drag&Drop operations and certain keystrokes, we quantify the KE's effort with the system. In a fixed interval (every 10 inputs) snapshots of the construction metrics (Status view) are saved to record the PKGs evolution over time. Additionally, memory consumption and time performance of certain system modules are monitored.

Each one-hour long interview between the knowledge engineer (KE) and an expert had the same setting. One fixed author of this paper took over the role of KE and met the expert in a virtual telephone conference. The KE shared the screen and presented the GUI of our system (see Section 3.6) where the expert's data was already loaded. After a brief introduction, the KE started to ask questions about files and folders by traversing through the file system. The explanations of the participant enabled the KE to model the expert's personal knowledge as discussed in our approach (Section 3). Whenever the AI made predictions, the expert was asked if they are correct or not and feedback was

Table 2: The seven questions from the questionnaire with the answers of the four experts and their average values.

| Question | E1 | E2 | E3 | E4 | Avg. & SD |
|---|---|---|---|---|---|
| Q1: *How many years have you been working with the data?* | 13 | 7 | 4 | 0 | $6 \pm 5.48$ |
| Q2: *How much do words in the file names reflect your language use (vocabulary) at work (scale: $1-10$)?* | 9 | 8 | 9 | 9 | $8.75 \pm 0.50$ |
| Q3: *Estimate how much your language use (vocabulary) at work is represented by the established tags (percentage).* | 50 | 15 | 10 | 10 | $21.25 \pm 19.3$ |
| Q4: *The established tags meaningfully reflect the language use (vocabulary) at your work (scale: $1-7$).* | 7 | 6 | 4 | 6 | $5.75 \pm 1.26$ |
| Q5: *The established tags are assigned to meaningful classes (scale: $1-7$).* | 6 | 7 | 6 | 7 | $6.50 \pm 0.58$ |
| Q6: *The established tags are meaningfully structured in a taxonomy (scale: $1-7$).* | 7 | 6 | 5 | 4 | $5.50 \pm 1.29$ |
| Q7: *The established tags meaningfully relate to each other (scale: $1-7$).* | 5 | 7 | 6 | 7 | $6.25 \pm 0.96$ |

entered accordingly. Every 10 minutes the KE reviewed the current construction state by opening the Status view and changed the focus on parts which needed more attention. After about 50 minutes the session ended and the remaining time was used to let the expert complete a questionnaire about the data source and the modeled knowledge graph. In the next section, we present the questionnaire and the results in detail as well as the data which was logged by our prototype during the interviews.

### 4.2   Interview Results

The questionnaire at the interview's end consists of seven questions (Q) which are presented in Table 2 together with the experts' answers (E), their average value and standard deviation (Avg. & SD). We stated the first question (Q1) to check how familiar the participants are with the data. The second question (Q2) was asked to figure out if the experts think that the given data actually contains work related words. While Q3 tries to give a rough estimation on the PKG's recall in percentage, Q4 gives an approximate measurement about its precision with regard to created named individuals[6] in the PKG. From the third question on, we are interested in the experts' opinions about the final result that was modeled during the interview. A seven-point Likert scale is used for our opinion-based questions ranging from 1 ("fully disagree") to 7 ("fully agree"). The remaining questions aim at the estimation of meaningfulness in the populated ontology (Q5) and taxonomic (Q6) as well as non-taxonomic relations (Q7).

---

[6] The questions refer to "established tags", since we presented tags in the GUI for the named individuals in the personal knowledge graph (PKG).

Besides qualitative data, we also captured quantitative data points during the interview which are presented in Table 3. Measurements are listed per row, while dataset-expert pairs are ordered in columns. After the number of resources in the PKG (#Resources) and the counts regarding the knowledge engineer's (KE) effort in the GUI, we list the number of true and false assertions[7] made by KE and AI in individual construction phases. Furthermore, we calculate the AI's accuracy by counting how often the expert agrees (true positive and true negative) with reviewed predictions. The section about Management of Named Individuals is further split into Unification and Ontology Population. While the management includes assertions about types, preferred/hidden labels and `foaf:topic`-relations, the latter two only consider `owl:sameAs` and ontology related assertions. Due to a software error in the taxonomy-module during the first two interviews, unfortunately, no broader concepts could be predicted. On the table's bottom all assertions by the KE (whether true or false) and all inputs (clicks, enter keys, drag&drop operations) are aggregated to calculate a assertions per inputs ratio. The Management of Named Individuals does not have an accuracy value (N/A), since each term automatically turns into a named individual and no suggestions for preferred and hidden label are made.

Since we continuously recorded measurements, we are able to examine the evolution of the PKG with respect to the inputs performed in the GUI. The development of the taxonomic and non-taxonomic part of the PKG is presented through several plots in Figure 4. We consider named individuals of type `skos:Concept` as taxonomy concepts (Figure 4a) and the remaining typed ones as non-taxonomic instances (Figure 4d). By looking at the number of graph components (Fig. 4b and 4e), one gets an idea of the connectedness over time. In addition, Figure 4c plots the number of concepts which are connected to at least one broader concept. Similarly, Figure 4f shows the average diameter (the greatest distance between any pair of instances) of non-taxonomic components to visualize the closeness among them.

The next section will discuss the results with regard to our research questions.

### 4.3 Discussion

Since file names are rather unusual sources to build PKGs from, we ask at the beginning of the paper the following question (RQ1): *Are file systems promising sources for knowledge graph construction?* Our experts agree that words they saw in the file names reflect their language use at work with an average value of 8.75 out of 10 (Q2 in Table 2). Having a higher-level management background, expert E4 came in daily work not in touch with file system F3 (see Q1 in Table 2), but was still able to recognize and explain the terms. Answers to questions Q4 to Q7 in our questionnaire (Table 2) indicate that we modeled all individual PKGs in a meaningful way for the experts. For these reasons, we conclude that file systems are promising sources for building PKGs.

---

[7] False assertions by AI mean that it later rejected initially true ones because of human feedback.

Table 3: Quantity of true and false assertions stated by the knowledge engineer (KE) and the AI for individual construction tasks. Additionally, the KE's GUI effort and the AI's accuracy is given.

| Measurement | SS1 (E1) | FS1 (E2) | FS2 (E3) | FS3 (E4) |
|---|---|---|---|---|
| #Resources | 88 | 50 | 39 | 32 |
| KE Clicks | 599 | 602 | 359 | 356 |
| KE Enter-Key | 60 | 56 | 30 | 47 |
| KE Drag&Drop | 26 | 34 | 21 | 18 |
| **Domain Terminology Extraction (Section 3.2)** | | | | |
| KE True | 82 | 50 | 33 | 26 |
| KE False | 48 | 44 | 14 | 72 |
| AI True | 400 | 270168 | 242149 | 948405 |
| AI False | 286 | 220285 | 106573 | 617366 |
| AI Accuracy | $0.67 = 45/67$ | $0.72 = 59/82$ | $0.83 = 35/42$ | $0.31 = 25/80$ |
| **Management of Named Individuals* (Section 3.3)** | | | | |
| KE True | 102 | 68 | 39 | 58 |
| KE False | 30 | 24 | 15 | 25 |
| AI True | 462 | 32161 | 8223 | 37159 |
| AI False | 4 | 1 | 23 | 155 |
| AI Accuracy | N/A | N/A | N/A | N/A |
| **Unification* (Section 3.3)** | | | | |
| KE True | 10 | 2 | 2 | 0 |
| KE False | 6 | 18 | 12 | 4 |
| AI True | 8 | 10 | 7 | 2 |
| AI False | 0 | 0 | 0 | 2 |
| AI Accuracy | $0.57 = 4/7$ | $0.10 = 1/10$ | $0.14 = 1/7$ | $0.00 = 0/2$ |
| **Ontology Population* (Section 3.3)** | | | | |
| KE True | 105 | 78 | 61 | 55 |
| KE False | 73 | 29 | 22 | 19 |
| AI True | 134 | 102 | 92 | 85 |
| AI False | 1 | 8 | 6 | 2 |
| AI Accuracy | $0.23 = 18/78$ | $0.65 = 30/46$ | $0.66 = 23/35$ | $0.48 = 12/25$ |
| **Taxonomy Creation (Section 3.4)** | | | | |
| KE True | 21 | 19 | 14 | 12 |
| KE False | 0 | 0 | 4 | 8 |
| AI True | N/A | N/A | 9 | 10 |
| AI False | N/A | N/A | 0 | 0 |
| AI Accuracy | N/A | N/A | $0.56 = 5/9$ | $0.20 = 2/10$ |
| **Non-Taxonomic Relation Learning (Section 3.5)** | | | | |
| KE True | 5 | 23 | 33 | 7 |
| KE False | 0 | 42 | 20 | 0 |
| AI True | 0 | 52 | 42 | 0 |
| AI False | 4 | 11 | 5 | 0 |
| AI Accuracy | 0/0 | $0.19 = 10/52$ | $0.52 = 22/42$ | 0/0 |
| **Aggregated** | | | | |
| All KE Assertions | 482 | 397 | 269 | 286 |
| All KE Inputs | 685 | 692 | 410 | 421 |
| KE Assertions/Inputs | 0.70 | 0.57 | 0.66 | 0.68 |

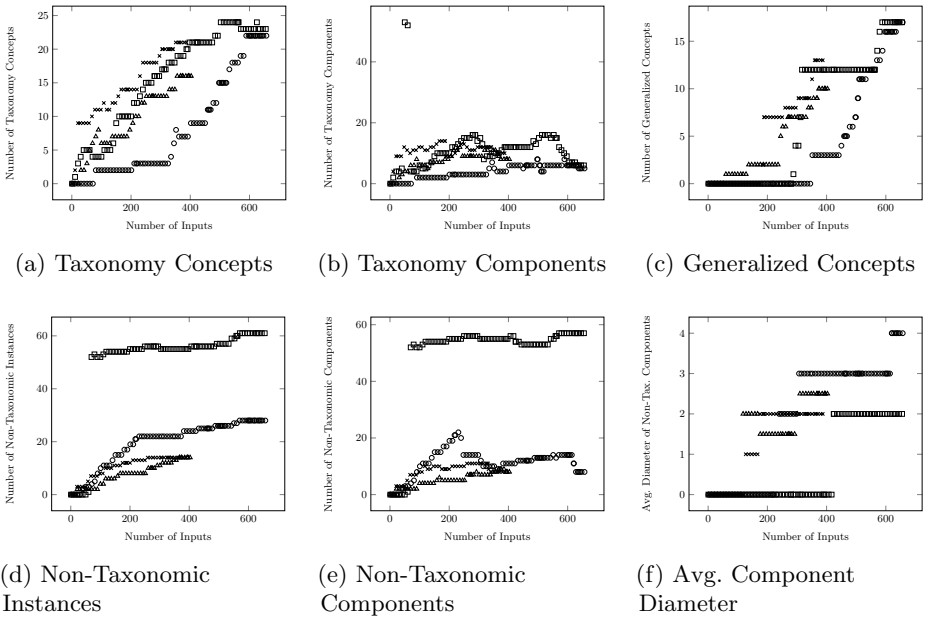

Fig. 4: Plots about the taxonomic and non-taxonomic parts of the PKG with respect to the number of inputs made in the GUI. For each dataset a symbol is assigned to recognize them: SS1 (□), FS1 (○), FS2 (×) and FS3 (△).

Because a completely manual construction can be time-consuming and thus AI could help in this process, we asked the next question (RQ2): *Can our system suggest helpful statements during usage?* In our approach, we consider the application of AI in several tasks ranging from (a) initial selection of domain relevant terms, (b) unification suggestions, (c) recommendation of class memberships, (d) suggestion of broader concepts and (e) prediction of non-taxonomic relations. How they performed can be obtained from Table 3 in form of accuracy values which calculate how often an expert agreed to suggestions stated by AI. (a) Since we do not consider multi-word terms in the extraction of domain relevant words, such terms had to be corrected frequently, which leads to a drop in performance. (b) Our unification rules tend to suggest more false positives leading to low accuracy scores, since they are designed with a high recall in mind. (c) The prediction of class assignments show mediocre results, since only preferred labels in combination with gazetteer lists are used to extract features. (d) For the taxonomy creation, our language resource GermaNet tended to suggest too general concepts which is why they were often considered unsuitable by our experts. (e) Regarding non-taxonomic relation learning, far to little examples were provided in case of SS1 and FS3 to be able to predict similar relations. All in all, there is a tendency that in certain cases helpful statements can be automatically suggested, but more research has to be done to further improve AI.

Concerned about the approach's practicability, we stated the third question (RQ3): *How efficient is the construction in our approach?* Effort measurements in Table 3 indicate that one input operation results in 0.6 to 0.7 assertions, thus already two inputs lead to a true or false statement. We assume that a value below 1.0 comes from not negligible GUI navigation and search efforts. Still, many clickable (bulk) feedback buttons combined with suggestions from the AI seem to yield to this positive outcome. Especially the Drag&Drop feature turns out to be a simple and fast way to relate resources to each other. Figure 4 visualizes how taxonomies and graphs evolve over entered inputs[8]. In comparison, the maintenance of taxonomies seem to require less effort than the non-taxonomic graphs, probably because only `skos:Concept`s and the `skos:broader`-relation need to be considered. The high diameter values of non-taxonomic graphs further indicate that resources in subgraphs are rather loosely connected. In summary, with moderately spent effort our KE was able to create, accept and also reject many assertions that eventually formed a meaningful personal knowledge graph. Still, efficiency could be further improved by better supporting the construction of the graph's non-taxonomic part.

## 5    Conclusion and Outlook

In this paper, we investigated the construction of personal knowledge graphs from file names with a human-in-the-loop approach. A case study with four independent expert interviews showed that the file system is a promising source, while suggestions by AI help to build such graphs with moderate effort.

Since we could not examine all of the aspects in detail, future work may further investigate in the challenges. For instance, there is potential for improvements in machine learning models, especially for the prediction of non-taxonomic relations. More sophisticated solutions could be applied in the extraction of domain terminology, including disambiguation and the discovery of multi-word terms.

**Acknowledgements** This work was funded by the BMBF project SensAI (grant no. 01IW20007).

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
