# OpenReview forum: "A Human-in-the-Loop Approach for Personal Knowledge Graph Construction from File Names"
_kg-construct.github.io/KGCW/2022/Workshop — KGCW 2022_

### Official Review · ~Jakub_Klímek1 · 2022-03-15
**Nice experiment and a clearly written paper**

**Rating:** 8
**Confidence:** 4

**Review:**

The authors report on an experiment where they construct a personal knowledge graph out of terms used in file names in a file system. The novelty of the approach is the fact that it is semi-automatic, involving an interview of the user with an expert, whereas other approaches rely on fully-automatic methods.

The paper is easy to read. The research questions, the approach, the experiment setup and the results are clearly presented and discussed. I also appreciate the screencast of the tool used for the approach.

Unfortunately, I was not able to run the program itself, I received an error.
```
PS D:\Downloads\kecs-main\kecs-main\dist> java -jar kecs.jar --mode Demo
Error: Invalid or corrupt jarfile kecs.jar
```
Edit 2022-03-18: this was my fault. I did not realize the repository was using git lfs. I was informed about this by the authors and indeed, the program works as described.

I have only one minor issue: Some references do not have DOIs (or access URLs) specified even if they do have them, e.g. [4] Dinneen, J.D., Julien, C.: The ubiquitous digital file: A review of file management research. J. Assoc. Inf. Sci. Technol. 71(1), E1–E32 (2020) - which in fact is https://doi.org/10.1002/asi.24222

---

### Official Review · ~Samaneh_Jozashoori1 · 2022-03-30
**An interesting semi-automatic approach to create personal knowledge graph using file names**

**Rating:** 8
**Confidence:** 3

**Review:**

This paper presents an interesting semi-automatic approach to create a personal knowledge graph using file names. I consider the emphasis of the approach on keeping the human in the loop while automatizing the creation of the knowledge graph, a positive point. The paper is well-written; each section is explained clearly in detail. The authors raise three research questions at the beginning of the paper which are tried to be answered by user experience evaluation explained in section 4. Links to the prototype, demo video and code are also provided in the paper.

minor issues and suggestions:
- There are four terms used by the authors in different sections that have brought confusion for me: "knowledge workers" e.g., in the abstract, "user" e.g., second paragraph of section 2, "knowledge engineer" e.g., section 3, and "Expert" e.g., section 4. At some points "knowledge engineer" and "expert" can be the same person because a knowledge engineer needs to be a domain expert in fact, or is the "user" or the "expert" the one who provides the "File system"? I would say as a reader I would have appreciated it if the minimum number of required terms were used in the paper and clear differentiation between them was provided as well.
- Figure 4 could be easier to follow if different colors or larger sizes were used.

---

### Official Review · ~Boris_Villazon-Terrazas1 · 2022-04-03
**A good starting point with a lot of potential**

**Rating:** 8
**Confidence:** 3

**Review:**

This paper investigates the construction of personal knowledge graphs from file names by including human experts in the loop.
The paper is well written, organized, and joyful to read. I really like the honesty of the authors, for example: "... shows mediocre results..."
Including knowledge experts is the way to go for building good knowledge graphs, I think your approach is coming back to the roots to Knowledge Representation. Finally, I also like you referenced the beginnings of the semantic desktop research, e.g., Leo Sauermann.

I have some minor questions
- how much data was used for training the random forest model? how much data was used for testing such model? I am not sure you present the final accuracy of such model.
- why did you not also consider jaro-winkler distance? ... it is good for "short texts", which I think is your case.

This work can be extended to a prototype that generates a data catalog / business glossary from datalake/blobstorage file systems in enterprises that have their data in some cloud vendor; do not underestimate such potential.

---

### Official Review · ~Sergio_José_Rodríguez_Méndez1 · 2022-04-04
**A well structured and presented paper along with an excellent prototype about a semi-automatic approach for personal knowledge graph construction from file names**

**Rating:** 8
**Confidence:** 3

**Review:**

## Overview:
The paper investigates the construction of personal knowledge graphs (PKGs) from file names with a human-in-the-loop approach (knowledge engineer -- KE).
A case study with four independent expert interviews showed that the file system is a promising source for PKGs, while suggestions by an "AI" help build such graphs with moderate effort.

## Positive remarks:
+ The paper is structured and presented very well.  Its readability is high.
+ The problem context (research questions), the proposed approach and methods, the conducted experiments, the results, and their analysis are easy to follow and clearly explained.
+ The prototype GUI and structure are well thought out: superb job!
+ The paper provides the links to all the prototype resources, including a demo video and the source code repo.

## Negative remarks:
No negative remarks.

## Comments:
- page 6: "random forest model [6]"; are you using the first algorithm created in 1995 by Ho or any recent variation?
- page 8: "overall assessment score": what is this score? How to calculate it?
- The authors should include a formal algorithm spec of one of the "AI" implemented logic, for example, the "Ontology Population" component described on page 6.  It would enhance the paper's readability.
- sec. 5: "More sophisticated solutions could be applied in the extraction of domain terminology"; however, most likely, one may need several different models/approaches for domain-specific named-entity recognition (NER) techniques, i.e. the "Computer Science" domain (see [1]) is different from the "Biomedical" domain.  I'd like to read the authors' comments about this.

[1] D'Souza, Auer. (2022). "Computer Science Named Entity Recognition in the Open Research Knowledge Graph".
http://arxiv.org/abs/2203.14579v1

## Minor corrections:
- page 6: SKOS, add a citation.
- change all words "web ____" to "Web ____" (with capital W): found only one on page 7.

---

### Decision · Program_Chairs · 2022-04-11

**Decision:**

Accept

**Comment:**

Dear authors,

Thank your for submitting your paper. We are happy to inform you that we accept your paper! Please carefully consider the reviews when you prepare your paper for the camera-ready version. You will receive specific instructions to submit your camera-ready soon.

Kind regards
Organizers of the Knowledge Graph Construction workshop 2022